# SARS-CoV-2 Spike Protein and Its Receptor Binding Domain Promote a Proinflammatory Activation Profile on Human Dendritic Cells

**DOI:** 10.3390/cells10123279

**Published:** 2021-11-23

**Authors:** Dante Barreda, César Santiago, Juan R. Rodríguez, José F. Rodríguez, José M. Casasnovas, Isabel Mérida, Antonia Ávila-Flores

**Affiliations:** 1Department of Immunology and Oncology, Spanish National Centre for Biotechnology, 28049 Madrid, Spain; dantebarlan@gmail.com; 2Department of Macromolecular Structures, Spanish National Centre for Biotechnology, 28049 Madrid, Spain; csantiag@cnb.csic.es (C.S.); jcasasnovas@cnb.csic.es (J.M.C.); 3Department of Molecular and Cell Biology, Spanish National Centre for Biotechnology, 28049 Madrid, Spain; jr.rodriguez@cnb.csic.es (J.R.R.); jfrodrig@cnb.csic.es (J.F.R.)

**Keywords:** COVID-19, ACE-2, inflammation, cytokines

## Abstract

Dendritic cells (DCs) are the most potent antigen-presenting cells, and their function is essential to configure adaptative immunity and avoid excessive inflammation. DCs are predicted to play a crucial role in the clinical evolution of the infection by the severe acute respiratory syndrome (SARS) coronavirus (CoV)-2. DCs interaction with the SARS-CoV-2 Spike protein, which mediates cell receptor binding and subsequent fusion of the viral particle with host cell, is a key step to induce effective immunity against this virus and in the S protein-based vaccination protocols. Here we evaluated human DCs in response to SARS-CoV-2 S protein, or to a fragment encompassing the receptor binding domain (RBD) challenge. Both proteins increased the expression of maturation markers, including MHC molecules and costimulatory receptors. DCs interaction with the SARS-CoV-2 S protein promotes activation of key signaling molecules involved in inflammation, including MAPK, AKT, STAT1, and NFκB, which correlates with the expression and secretion of distinctive proinflammatory cytokines. Differences in the expression of ACE2 along the differentiation of human monocytes to mature DCs and inter-donor were found. Our results show that SARS-CoV-2 S protein promotes inflammatory response and provides molecular links between individual variations and the degree of response against this virus.

## 1. Introduction

Dendritic cells (DCs) are bone-marrow-derived cells distributed at the main entry sites of pathogens as immature dendritic cells (iDCs). These antigen-presenting cells (APCs) express a vast arsenal of pattern recognition receptors (PRRs), through which they sense a broad range of pathogens-associated molecular patterns (PAMPs) for its capture and processing [1]. The encounter with the antigen turns on a maturation process, which leads them to the acquisition of migratory capacity and expression of MHC–peptide complexes and co-stimulatory molecules. Mature dendritic cells (mDCs) reach the lymph node and present the processed antigens to naïve T cells at the immune synapse (IS) context. DCs are the bridge between innate and adaptive immunity since they also modulate the extent and nature of the T cell response by secreting large amounts of cytokines in response to PRR ligation [1,2,3]. DCs essential activity must be tightly regulated to achieve an effective T cell response while avoiding exacerbating inflammation. Deepening into the mechanisms of immune system activation and regulation by DCs will provide valuable tools for counteracting pathogens and designing effective vaccines.

DCs are key components in viral infections, either through type I interferon production and the establishment of the antiviral state, or by directly activating the adaptive cellular immune response [4,5]. Their correct function is critical in the control and clinical evolution of the infection by the severe acute respiratory syndrome (SARS) coronavirus (CoV)-2 [6,7]. This virus is the etiologic agent of COVID-19, the ongoing pandemic currently affecting the global community [8,9]. SARS-CoV-2, as the SARS-CoV virus, infects cells via binding of its envelope Spike glycoprotein (S protein) with the angiotensin-converting enzyme 2 (ACE2), a carboxypeptidase abundantly expressed in type I and II alveolar epithelial cells and also present in the cell surface of many cells [10]. S protein includes two domains. The N terminal region or S1 harbors the receptor-binding domain (RBD), which is responsible for ACE2 binding. The C terminal region or S2 mediates the fusion of the viral particle with the host cell [11].

In most cases, the immune system can control and resolve SARS-CoV-2 infection, and patients experience no symptoms or mild to moderate illness with prompt recovery. Nevertheless, nearly 1–5% of the population, particularly the eldest, develop serious illness, and in some cases death [7,12]. Morbidity and mortality of COVID-19 are closely linked to a dysregulation of the immune response [13]. Lung analysis of seriously ill COVID-19 patients reveals pathogenic inflammatory activation, associated with cytokine storm, whereas blood analyses show severe lymphopenia. Such dysregulation of the immune response has been associated with an aberrant response of the innate immune system [14]. Recent studies point out that genetic variability impacts the activity of innate immune cells, which might explain the high variability of the severity of the disease. Other factors that contribute to the fluctuation of the innate immune response are aging and pre-existing diseases [15,16].

Information regarding the expression of the canonical SARS-CoV-2 receptor in DCs is limited and contradictory [17,18]. Few studies have shown that DCs can be infected by SARS-CoV-2, although such infection is not productive. Here we show that human DCs express ACE2 and analyze their response to SARS-CoV-2 S protein or RBD challenge. Both viral proteins increased the expression of maturation markers, including MHC and costimulatory molecules. The analysis of activation of key signaling molecules involved in inflammation demonstrates a close correlation between activating signaling intermediates and the expression and secretion of proinflammatory cytokines. Differences in the expression of ACE2 along the differentiation of human monocytes to mature DCs and inter-donor were found. DCs interaction with SARS-CoV-2 S protein is extremely relevant considering the current wide use of S protein-based vaccination protocols [19]. DCs maturation upon S protein binding is expected to be a key step to induce long-lasting immunity against this virus. On the contrary, excessive dysregulated production of inflammatory cytokines may provide molecular links between individual variations and possible dysregulation of the immune response.

## 2. Materials and Methods

### 2.1. Cell Culture, Stimulation, and Blocking Assays

Peripheral blood mononuclear cells (PBMC) were obtained by Lymphoprep^TM^ (Cat# 1114545, Alere Technologies, Oslo, Norway) density gradient from buffy coats of healthy human donors (Community of Madrid Transfusion Center). CD14^+^ cells were purified using human anti-CD14-labeled magnetic beads and LS columns (Cat# 130-050-201 and 130-042-401, Miltenyi Biotec, Auburn, CA, USA). Eluted cells were labeled with anti-human antibody CD14-FITC (Cat# IM065, clone RM052, Beckman Coulter, Brea, CA, USA) to determine yield and purity, which was of a least 96% (Appendix A). Cells were plated (1 × 10^6^ cells/mL) in RPMI 1640 medium (Biowest, Riverside, MO, USA) supplemented with 10% fetal calf serum (Gibco, ThermoFisher Scientific, Waltham, MA, USA), 100 U/mL penicillin, 100 µg/mL streptomycin, and 200 mM L-Glutamine (Sigma-Aldrich, St. Louis, MO, USA) (Complete medium). To generate monocyte-derived dendritic cells, the medium was supplemented with 50 ng/mL of human granulocyte and monocyte colony-stimulating factor (GM-CSF) and 25 ng/mL of human interleukin-4 (IL-4) (Cat# 21-8339 and Cat# 21-8044, Tonbo, San Diego, CA, USA) at days 0 and 3; and incubated at 37 °C and 5% of CO_2_. At day 6, immature dendritic cells (iDCs) were harvested. 

For stimulation, iDCs were harvested and plated (1.66 × 10^6^ cells/mL) in fresh complete medium containing GM-CSF and IL-4. Spike protein (10 µg/mL), the RBD (10 µg/mL) or LPS (0.1 µg/mL) (Sigma-Aldrich, St. Louis, MO, USA) as control, were added and cells were incubated as indicated. Twenty-four hours post stimulus, cells were harvested and supernatants collected. Maturation cocktail was added (10 ng/mL TNF-α, 10 ng/mL IL-1β, and 1 μg/mL PGE2, Biolegend, San Diego, CA, USA) to iDCs, and 24 h later mature dendritic cells (mDCs) were harvested.

Blocking assay was performed incubating iDCs (1 × 10^6^/mL) in complete medium with anti-DC-SIGN antibody (40 µg/mL) (Cat# MAB16211, RD Systems) at 37 °C, for 30 min. After two washes, cells were stimulated as indicated during 24 h and then analyzed.

At the end of the different treatments, harvested cells were washed twice and centrifugated at low speed (150× *g*, 10 min) to discard death cells. Cell pellets were suspended, and cell number and cell viability were determined by using Trypan Blue Exclusion in a Life Countess ii (ThermoFisher Scientific, Waltham, MA, USA). In general, DCs cultures have low dead index, with a cell viability above 85%.

### 2.2. SARS-CoV-2 Spike and RBD Proteins Production

Purified Spike protein was prepared as described in [20]. Briefly, a recombinant cDNA cloned in pCDNA3.1 vector and coding for soluble S protein containing a T4 fibritin trimerization sequence, a Flag epitope and an 8xHistag (all at the C-terminus), was transfected in HEK293F cells using standard procedures. Protein was purified by Ni-NTA affinity chromatography from cell supernatants and transferred to 25 mM Hepes, pH 7.5; and 150 mM NaCl, during concentration. To produce SARS-CoV-2 S RBD, a gene optimized for insect cell expression encoding the RBD (residues 319 to 541) from the Wuhan SARS-CoV-2 strain Spike protein (GenBank: MN908947.3) was used. The gene contained the Spike leader sequence (residues 1 to 14), and a hexa-histidine tag at its N- and C-termini, respectively, was synthesized and cloned (Genscript, Piscataway, NJ, USA) into the pFastBac1 expression vector (Thermo Fisher Scientific). The resulting plasmid was used to generate the recombinant baculovirus rBV_RBD_his using the Bac-to-Bac technology (Thermo Fisher Scientific, Waltham, MA, USA). The recombinant RBD protein was produced by infecting HighFive cells (Thermo Fisher Scientific, Waltham, MA, USA) with rBV_RBD at a multiplicity of infection of three plaque forming units per cell. Infected cultures were maintained in TC-100 medium (Thermo Fisher Scientific, Waltham, MA, USA) for 72 h. After this period, cell medium was harvested and clarified by centrifugation (4500× *g* for 10 min) and filtration through 0.45 µm filters. Protein purification was achieved by immobilized metal affinity chromatography (IMAC) followed by gel filtration. IMAC was carried out using 5 mL nickel NTA agarose cartridges (Agarose Bead Technologies S.L., Doral, FL, USA) at a flow rate of 1.5 mL/min. Retained protein was eluted with a linear gradient of 500 mM Imidazole in Tris-saline buffer (pH 7.5). Fractions were analyzed by SDS-PAGE, and those containing the RBD polypeptide were pooled together and concentrated using Amicon Ultra-15 centrifugal units with a 10-kDa cutoff membrane (Millipore, Burlington, MA, USA). The concentrated protein was loaded onto a Superdex 75 10/300 Increase gel filtration (GE Healthcare, North Richland Hills, TX, USA) equilibrated with PBS. RBD peak fractions were analyzed by SDS-PAGE and pooled together. Purified protein aliquots were maintained at −20 °C.

### 2.3. Flow Cytometry

Cells were collected and stained (20 min, RT) with anti-human antibodies CD11c-PE (Cat# 1760, clone BU15), CD40-PE (Cat# 1636, clone mAb89), APC-CD80 (Cat# B30642, clone MAB104), CD83-FITC (Cat# IM2410, clone HB15a), CD86-PE (Cat# IM2729, clone HA5.2B7), HLA-ABC-FITC (Cat# IM1838U, clone B9.12.1) and HLA-DR-FITC (Cat# 1638, clone Immu-357) all from Beckman Coulter, Brea, CA, USA. 

Cells were fixed (10 min, RT) with 1% p-formaldehyde (PFA) in phosphate-buffered saline (PBS: 10 mM sodium phosphate, 0.15 M sodium chloride, pH 7.2). Cells were washed and resuspended in 200 µL of PFA 1%. Data was acquired (at least 30,000 events per sample in DC gate) on a Cytomics^TM^ FC 500 (Beckman-Coulter, Brea, CA, USA) flow cytometer, and the analysis were performed using FlowJo software version 10.2, FlowJo (LLC, Ashland, OR, USA).

DCs cultures have very low dead index and high cell viability. Flow cytometry further analysis was done by selection of singlets followed by FSC and SSC gating strategy to discard debris and identify DCs population (Appendix A). The fluorophore-conjugated antibodies were combined in three staining mixes to analyze the different molecules. Unstained cells, single stained, and cells fluorescence minus one (FMO) condition were processed and acquired in parallel to identify background levels of staining (Appendix A).

For ACE2 detection, anti-ACE2 (Cat# 15348, Abcam, UK) primary antibody and a secondary anti-Rabbit IgG-FITC (Cat# 4041-02, SouthernBiotech, Lemere, CA, USA) were used. Unstained cells and cells stained with the secondary antibody were used as controls.

### 2.4. Quantitative Reverse Transcriptase Polymerase Chain Reaction (qRT-PCR)

Twenty-four hours after stimulation, cells were collected, and mRNA was extracted using TRIzol^®^ (Invitrogen, Waltham, MA, USA) reagent according to the manufacturer’s instructions. The RNA was resuspended in 30 µL of nuclease-free water and quantified by Nanodrop. One microgram of RNA was used for first-strand cDNA synthesis using the SuperScript III Reverse Transcription Kit (Cat# 12574026, Thermofisher Scientific, Waltham, MA, USA). Specific probes were used for detection of IL-6 (F: 5′-GCTGAAAAAGATGGATGCTT-3′; R: 5′-GGCTTGTTCCTCACTACTCTC-3′), IL-1B (F: 5′-CTCGCCAGTGAAATGATGGCT-3′; R: 5′-GTCGGAGATTCGTAGCTGGAT-3′), IL-12 (F: 5′-CTCTGGCAAAACCCTGACC-3′;R: 5′-GCTTAGAACCTCGCCTCCTT-3′), TNF-α (F: 5′-TCAGATCATCTTCTCGAACCCC-3′; R: 5′-ATCTCTCAGCTCCACGCCAT-3′), IL-10 (F: 5′-GCC TAA CAT GCT TCG AGA TC-3′; R: 5′-TGA TGT CTG GGT CTT GGT TC-3′), IFNα (F: 5′-ATTTCTGCTCTGACAACCTC-3′; R: 5′-TGACAGAGACTCCCCTGATG-3′), IFN β (F: 5′-TGTGGCAATTGAATGGGAGGCTTGA-3′; R: 5′-TCAATGCGGCGTCCTCCTTCTG-3′) and GAPDH (F: 5′-CGACTTCAACAGCAACTCCCACTCTTCC-3′; R: 5′-TGGGTGGTCCAGGGTTTCTTACTCCTT -3′) as reference gene. The quantitative real-time PCR was performed on a StepOne system (Aplied Biosystem, Thermofisher Scientific, Waltham, MA, USA). Then, 2ΔΔCt method was used to determine the relative expression of each gene.

### 2.5. Enzyme-Linked Immunosorbent Assay (ELISA)

Cell culture supernatants were collected and stored at −70 °C until analysis. Concentration of IL-1β, IL-6, IL-10, and TNF-α was measured using the sandwich-type immunoassay ELISA MAX^TM^ Deluxe Set (BioLegend, San Diego, CA, USA) for each molecule. Briefly, 96-well MAXISORP microplates (ThermoFisher Scientific, Waltham, MA, USA) were coated with anti-human IL-1β, anti-human IL-6, anti-human IL-10, or anti-human TNF-α capture antibody, and then blocked. Diluted supernatants were added (1–2 h, RT). After four washes, each protein was recognized with its specific biotinylated detection antibody. Then, avidin-HRP and tetramethylbenzidine (TMB) were added. Reactions were read with a microplate reader spectrophotometer (MULTISKAN GO, ThermoScientific, Waltham, MA, USA).

### 2.6. Early Signaling Assay and Western Blot Analysis

iDCs were stimulated with purified SARS-CoV2 S protein or RBD according to protocols described with the SARS-CoV1 proteins [21]. Briefly, iDCs (3 × 10^6^) in complete medium were treated with the purified proteins (10 μg/mL) or LPS (0.1 μg/mL) for the times indicated. Cells were pelleted and suspended in ice-cold p70 buffer containing protease inhibitors, as described in [22]. Cell lysates were resolved by SDS/polyacrylamide electrophoresis and gels were blotted onto nitrocellulose membranes. These were blocked with 5% BSA in TBS (10 mM Tris-HCI, pH 8.0; 150 mM NaCl) and incubated with primary antibodies diluted in TBST (TBS plus 0.05% Tween X-100) containing 5% BSA (overnight, 4 °C). We used anti -phospho-STAT 1 Ser 727 (9177), -phospho-AKT Ser473 (4060), -AKT (2920), -phospho-MAPK Thr 202, Y204 (ERK1/2; 4370), -ERK1/2 (9107), -IκB (9242), all from Cell Signaling, Danvers, MA, USA; anti-tubulin mAb (T5168) was from Sigma-Aldrich, Sant Louis, MO, USA.

Blots were then incubated with fluorescent secondary antibodies anti-rabbit IgG Dylight 800 (SA5-35571; Invitrogen, Waltham, MA, USA) and anti-mouse IgG AlexaFluor 680 (175775, AbcamLife, Cambridge, UK) (1 h, RT), and visualized with an Odyssey scanner (LI-COR, Lincoln, NE, USA). Images were analyzed with ImageJ software (NIH, Bethesda, MD, USA; version 2.1.0/1.53c; 2010–2021). To determine loading amounts or degree of phosphorylation, densitometric values of each phospho-protein or protein were divided by that of the total protein or the loading control Tubulin. Normalized values were determined by considering the basal condition = 1. 

### 2.7. Immunofluorescence and Confocal Microscopy

2.5 × 10^5^ cells were attached to coverslips during 20 min then fixed with methanol and blocked (10% FBS, 3% BSA in PBS). After three washes, coverslips were incubated with anti-ACE2 primary antibody (10 µg/mL; 1 h, RT), and then with secondary antibody IgG anti-Rabbit-Cyanine 2 (dilution 1:100; Cat# 111-226-003, Jackson ImmunoResearch, West Grove, PA, USA; 1 h, RT). Cells were incubated with DAPI (dilution 1:1000; Cat# D1306, TermoFisher, Waltham, MA, USA; 10 min, RT). Coverslips were mounted with ProLong Gold Antifade Mountant (Cat# P36930, Invitrogen, Waltham, CA, USA). Images were acquired with an Olympus Fluoview FV-1000 confocal microscope. For mean intensity fluorescence (MFI) quantification, Z stacks images were captured from at least three donors, and at least 30 cells for each condition were analyzed. Quantification was performed using ImageJ software (NIH). For Z stack images, 10 sequential planes were acquired at axial (z) spacing each 0.5 µm to form a z-stack. The imageJ channel corresponding to ACE2 signal was flattened along the z axis as maximum intensity projections representing a “top” (x-y) view of the volume. Region of interest (ROI) was set for every individual cell and MFI determined using the ROI manager tool.

### 2.8. Statistics

Analyses were performed using GraphPad Prism version 6 software (San Diego, CA, USA). Data are shown as mean ± SEM. Normality was assessed using the D’Agostino-Pearson test. Student’s *t*-test or ANOVA and Tukey’s multiple comparisons test were used and indicated for each case in the corresponding figure legend. A value of *p* ≤ 0.05 was considered significant.

## 3. Results

### 3.1. SARS-CoV-2 Spike Protein and Its RBD Promote Maturation and Activation of DCs

When iDCs recognize antigens, they turn on a maturation program characterized by cytokine secretion, as well as expression of surface molecules that led to the acquisition of migratory and costimulatory capacities. DCs thus acquire features that warrant their function as APC. To investigate how human DCs respond against the S protein of SARS-CoV-2, we differentiated primary DCs from monocytes of healthy donors and challenged them with either complete S protein or with the region that contains the RBD region. Treatment with Lipopolysaccharides (LPS) was included as control to evaluate the state and responsivity of the iDCs.

DC maturation phenotype was first evaluated by measuring CD11c and MHC-II expression by flow cytometry (Figure 1). As expected for our model, most cells expressed both molecules (Figure 1A, +90% of the cells). However, bi-parametric analysis of these markers showed increased percentage of DCs expressing high levels of MHC-II (MHC-II^high^) in the presence of S protein or RBD stimuli. S protein and its binding domain induced MHC-II^high^ percentages similar to that induced in the LPS stimulated cells (Figure 1B,C). As in the case of MHC-II, most of the DCs also expressed MHC-I. All the stimulus promoted increases in the number of cells expressing high levels of the marker (MHC-I^high^; Figure 1D). RBD showed a stronger effect than S protein for MHC-I up-regulation, promoting a percentage of expression similar to the positive control (Figure 1D). CD83 is the canonical DC phenotypic marker that indicates maturation in response to antigen recognition [23]. Different from MHC molecules, non-stimulated iDC expressed low levels of this marker. We observed a sharp increase in the number of cells expressing high levels of CD83 following stimulation, with RBD displaying an up-regulation similar to that exerted by LPS. Increases in the amount of CD83 expressed per cell were also noticeable (Figure 1E).

During their activation and maturation, DCs up-regulate the expression of adhesion molecules necessary for cell migration, and in vitro maturation of DC correlates with clusters formation [24]. S protein and RBD-challenged iDCs showed cluster formation, while DCs unstimulated (iDCs) do not form them (Figure 2A), suggesting up-regulation of adhesion molecules in response to these viral proteins. Known as the second signal, the expression of co-stimulatory molecules in DCs during antigen presentation is required for proper and complete activation of naïve T cells. mDCs augment the levels of co-stimulatory molecules on the cell surface [25]. We analyzed the expression of the co-stimulatory molecules CD40, CD80, and CD86 in S and RBD-stimulated DCs 24 h post-stimulus (Figure 2B–D). Flow cytometry analysis shows that S and RBD elicit up-regulation of all the co-stimulatory molecules analyzed. iDCs expressed CD40, but those treated with S protein or RBD augment the expression of this molecule (Figure 2B). Both viral proteins also upregulated the number of cells expressing CD80 and CD86, but only RBD promoted a significantly augmented amount of these markers expressed per cell (Figure 2C,D).

Together our analysis indicates that S protein and RBD proteins induce DCs maturation, with clear up-regulation in the expression of MHC and costimulatory molecules. The effect of RBD was stronger than S in all the molecules analyzed and was statistically comparable to those exerted by LPS on DCs.

### 3.2. SARS-CoV-2 Spike Protein and Its RBD Promote a Proinflammatory Activation Profile in DCs

DCs activation and maturation upon antigen recognition promote selective cytokine production, which depends on the antigenic nature of the pathogen. Cytokine production for DCs is pivotal for naïve T cell activation and polarization. To investigate DCs activation profile induced by S protein and RBD, we evaluated a set of cytokines at mRNA level by quantitative RT-qPCR. IL-6, TNFα, IL-1β, IL-12, IL-10, IFNα, and IFNβ were measured at 12 and 24 h post-stimulation (Figure 3). Since all the experiments were performed using cells from different healthy donors, highly variable responses between the donors were expected. This variable response between donor was also observed when the expression at 12 h post-stimulation was analyzed (Appendix A). The highest augments of cytokines due to S protein and RBD stimulation were observed for proinflammatory cytokines (Figure 3A–D and Appendix A). For some donors, RBD induced an expression of IL-6, TNFα, and IL-1β comparable to LPS treatment (Appendix A).

To validate the proinflammatory profile triggered in DCs due to S protein and RBD treatment, we measured cytokines in supernatants of S protein and RBD-treated DCs 24 h post-challenge. LPS is a canonical PAMP that exerts a proinflammatory program, and as expected, in our model, LPS elicited an extensive production of IL-6, TNFα, IL-1β, and IL-10 (Figure 4). No significant augments of cytokines were observed in supernatants of S protein-stimulated DCs (Figure 4). Differently, high amounts of IL-6, TNFα, and IL-1β were produced by RBD-treated iDCs (Figure 4A–C). A tendency to produce IL-10 was also observed in these cells, although this was not statistically significant (Figure 4D). Concentrations of IL-6 and TNFα at 12 h post stimuli were also measured (Appendix A). We documented a tendency to a cumulative increase of IL-6 secretion due to S protein and RBD treatment. Differently, in the case of TNFα, a higher concentration was detected at 12 compared to 24 h, which might be indicative of auto-consumption of this cytokine by S protein and RBD-treated DCs. Our results indicate that RBD, and to a lesser extent S protein, were recognized by DCs and promoted a proinflammatory activation profile.

### 3.3. SARS-CoV-2 Spike Protein and Its RBD Trigger a Proinflammatory Signaling on DCs

iDC activation and maturation processes are governed by intracellular complex signaling pathways that are becoming untangled. To explore the signaling events triggered by S protein and RBD recognition by iDCs, we analyzed the phosphorylation state of different key signaling proteins. LPS was used as the control of specific TLR4 signaling and to validate the functional state of iDCs. This endotoxin promoted strong and sustained phosphorylation of STAT1, AKT, and ERK, as well as marked and sustained degradation of IκB, the first step in the activation of NFκB regulated transcription. S protein and RBD induced transitory STAT1 phosphorylation that peaked at 30 min and then reached its basal levels after 2 hours of stimulation (Figure 5). In contrast, S protein and RBD promoted sustained phosphorylation of the kinases AKT and ERK, as well as IκB degradation, with RBD displaying a higher effect than S protein (Figure 5).

Together our data indicated that RBD, and to a minor extent S protein, interactions with iDC trigger the signaling pathways known to promote the secretion of proinflammatory cytokines, as well as cell maturation and activation. The effects of the viral proteins are similar, but minor, to the activation triggered by LPS, which coincides with the gradation of the effects observed in the expression of the different maturation surface markers and cytokines secreted.

### 3.4. iDC-ACE2 Expression Varies among Different Donors

DCs sense and capture antigens by several mechanisms, including non-specific pinocytosis as well as specific receptor recognition. ACE2 is the canonical receptor for SARS-CoV-2, and the Spike-ACE2 interaction has been demonstrated [26]. A logical assumption is that either S protein or RBD protein directly binds to ACE2 in iDC, which would promote signaling events that lead to iDC activation and maturation. We thus evaluated ACE2 expression by immunofluorescence; through in vitro differentiation, we obtained mDCs from human monocytes. ACE2 was detected on monocytes, iDCs. and mDCs; in all these cell types, the enzyme showed both, a surface and intracellular localization (Figure 6A). Interestingly, the percentage of cells that express ACE2 seems to decrease along the process of differentiation; all monocytes expressed ACE2, while just 14% of iDCs and 9% of mDCs expressed this receptor (Figure 6A,B). When different donors were analyzed, significant variations in the amount of ACE2 expressed per cell was observed in iDCs. Nonetheless, such variations were not significant when the expression was analyzed in mDCs (Figure 6C). Expression of ACE2 in iDCs was also evaluated by flow cytometry (Figure 6D). We observed a similar percentage of ACE2 expression between donors, with an average of 10% (Figure 6E).

Together these analyses show that ACE2 expression is regulated along with the differentiation of DCs and that there are differences among individuals that might condition the amount of ACE2 levels in iDCs. This observation points to a possible variable response to S protein or RBD recognition if this occurs only through ACE2.

### 3.5. Dendritic Cell Activation in Response to Spike Protein and the RBD Is DC-SIGN Independent

DC-SIGN ligation in DC is reported to elicit ERK and PI3K activation [27]. Since we observed robust activation of these kinases by S protein and RBD, we next evaluated the expression of this lectin by Western blot and immunofluorescence in our model. As expected, iDCs expressed high amounts of the C-type lectin DC-SIGN (Appendix A). Both S protein and RBD proteins we used here are produced as glycosylated proteins, which opens the possibility that they might bind DC-SIGN. We assessed this possibility by evaluating the maturation phenotype of iDCs stimulated with S protein and RBD in the presence of DC-SIGN-blocking antibodies. DC-SIGN blockade did not interfere with the effects promoted by these proteins; CD83 and CD86 were expressed to a similar extent in control and DC-SIGN-blocked iDCs treated with S protein or RBD (Figure 7 and Appendix A). These results suggest that this lectin is not crucial in iDCs activation by S protein or RBD recognition.

## 4. Discussion

The binding of S protein to ACE2 is the main known mechanism for SARS-CoV-2 cell infection. According to this knowledge, currently available vaccines are primarily aimed to promote neutralizing antibodies production against S protein [28]. While several of these vaccines have proved to avoid severe disease symptoms, evidence suggests that antibody production is insufficient for long-lasting immunity and that eliciting T cell-mediated immunity will be desired to boost vaccines’ efficacy [29]. To this end, DCs must present antigens to naïve T cells in an MHC context. Nonetheless, information regarding the role of DCs in the SARS-CoV-2 vaccination context is still limited. Using an in vitro model, we analyzed the response of human iDCs to S protein and its RBD, and we found that both viral proteins activate DCs, with RBD triggering a strong proinflammatory response.

iDCs contain a wide range of PRRs through which they sense antigens. In response to antigen recognition, DCs turn on a maturation program characterized by a change in phenotypic markers expression. S protein and RBD promote DCs maturation as shown by enhanced expression of MHC class I and II, which is in agreement with the phenotypic profile of mDCs [3]. While both S protein and RBD increased the percentage of MHC-II-expressing cells, only RBD treatment induced a substantial increase in MHC-I expression, suggesting a different form of antigen processing. Both viral proteins also induced expression of CD83, a classical marker of maturation [30]. Up-regulation of co-stimulatory molecules is a hallmark of DCs activation [31]; both proteins induced CD40 and CD80 up-regulation, while RBD also upregulated CD86 expression. Overall, these findings demonstrate that DCs mature and activate due to S protein and RBD recognition. Since maturation is a consequence of antigen recognition, it is straightforward to assume that S protein and RBD were sensed by DCs.

Along with these experiments, we consistently observed that RBD-induced maturation was stronger than S protein. In concordance with levels of maturation markers, RBD elicits stronger induction of the RNA messengers of inflammatory cytokines compared to the S protein. Specifically, we observed high levels of IL-6, TNFα, and IL-1β, with a highly variable response between donors, a common feature of experiments performed using cells from human individuals [32]. The variations observed in the responses to S protein or its RBD domain could be due to differences in molecular weight, size, and tridimensional conformation of both antigens. Considering that we use the same concentration of both antigens, the higher molar ratio of RBD than S protein also could contribute to our results.

The variations we observed in the response to S protein and its RBD domain are in part coincident with that recently reported by Colunga et al. [33]. In this study, the effects triggered by inoculation of the S protein, or a truncated form that harbors the RBD domain, were evaluated in transgenic hACE2 mice. Whereas the truncated protein promoted acute lung injury, very likely through triggering massive inflammation, the complete Spike protein had little effect. This points to the potent inflammatory effect triggered by this region of the Spike protein. Discrepancies in the effects promoted by the S protein may be likely due to the different experimental models. While Colunga and coauthors use a complete animal, we used DCs. In transgenic hACE2 mice, it is expected that the epithelial hACE2-expressing cells exert a pivotal role, and different DCs possess several other PRRs apart from ACE2 that could increase the response to complete S protein.

ACE2 expression in DCs suggests a direct interaction between S protein or its RBD and ACE2 in this cell type. S protein requires a proteolytic step for RBD exposure and viral entry [10]; the requirement for an extra proteolytic step could also explain the greatest effect of the RBD. Additionally, the differences observed in response to stimuli may be related to the participation of different PRRs. In silico studies predicted TLR4 involvement in SARS-CoV-2 recognition [34,35]. Accordingly, some donors respond to RBD in a similar degree to LPS, suggesting that the activation pathway triggered by RBD is similar to that of LPS. Furthermore, cytokines produced by RBD-stimulated DCs follow a similar pattern to those produced by LPS-stimulated DCs. In agreement with our results, high levels of IL-6 and TNFα were detected in sera from SARS-CoV-2 patients [36,37]. Differently, no cytokines were detected in supernatants of DCs stimulated by S protein, in line with a few levels of activation documented at the mRNA level. The analysis of cytokine production shows that while IL-6 production tends to accumulate in response to RBD treatment, TNFα secretion appears to be higher at 12 than 24 h. TNFα contributes to an efficient DCs maturation during virus infection [38], suggesting that DCs in culture can consume self-produced TNFα, which could partially explain this observation. Further, the higher expression of maturation phenotypic markers in RBD-stimulated DCs could be influenced by TNFα.

The analysis of signaling pathways activated in response to viral proteins demonstrates that RBD, and to a minor extent S protein, trigger AKT and ERK activation, which reproduce the downstream pathway signaling upon SARS-CoV-2 infection in Vero cells [39]. DCs also showed slight IκB degradation compared to that observed with LPS, in correlation with the reduced cytokine production. Activation of STAT1 is pivotal for DCs maturation and a key modulator of type I IFN response, which plays a critical role in SARS-CoV clearance [40,41]. Recombinant S protein and RBD promotes STAT1 phosphorylation, suggesting a type I IFN response establishment. S protein possesses a high number of glycosylations, which have an important role in SARS-CoV-2 recognition [42,43]. On the other side, DCs possess several pattern-recognizing C-type lectin receptors, and among them, they express DC-SIGN, which binds several glycans on S protein, promoting a proinflammatory response and viral entry [44,45,46]. Our signaling experiments could represent activation via DC-SIGN recognition, but DC-SIGN blocking had no effects on DCs stimulation with S protein and RBD. The fact that other several pattern-recognizing C-type lectin receptors are capable to recognize glycans of S protein could explain that blocking DC-SIGN alone did not affect DCs activation [43]. 

Our experiments show ACE2 expression in DCs although the percent of positivity expression is low with an average of 14% in iDCs. This agrees with a recent observation by Hsu et al. showing that interaction between S protein and ACE2 elicits a hyperinflammatory response [47]. A low expression of ACE2 in human DCs, together with the observation that low levels of ACE2 limits infection [48], could partially explain a not productive viral replication in DCs. IF experiments suggest a higher percentage of ACE2-positive iDCs than mDCs. The contribution of angiotensin II (Ang-II) to DC differentiation and the differential regulation of Ang-II receptor and ACE2 [49,50] could explain our observations. Interestingly, intradonor ACE2 expression per cell is similar between iDCs and mDCs. Several studies correlate ACE2 expression levels with the severity of clinical features [51,52,53,54]. The donor variability observed for ACE2 expression on iDCs could facilitate differential responses elicited by the S protein or its RBD. Further work must address an accurate measurement of ACE2 molecule on the DCs surface to evaluate its role as a biomarker to predict possible dysregulation of the immune response and severity of the disease.

To sum up, we described here that RBD strongly promotes the activation and maturation of iDCs, which was coincident with the activation of key signaling molecules and cytokines that mediate inflammation. This supports the use of RBD in the design of efficiency-based vaccination strategies.

## Figures and Tables

**Figure 1 cells-10-03279-f001:**
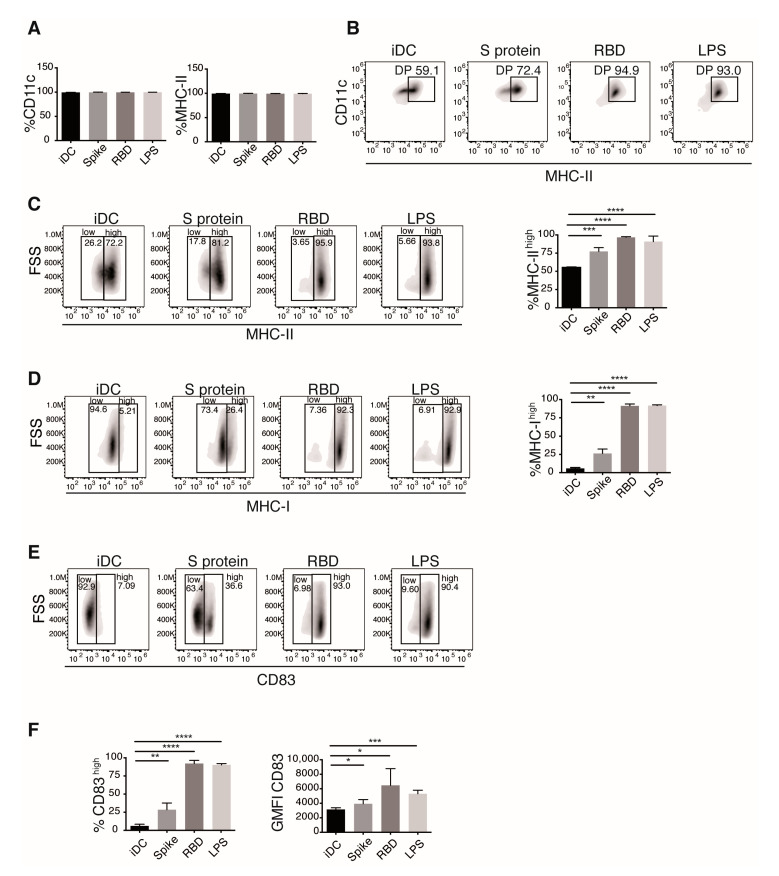
Spike protein and the RBD of SARS-CoV-2 promote maturation of iDCs. Cells were stimulated with Spike protein or RBD of SARS-CoV-2 or LPS as control for 24 h and then stained and analyzed by flow cytometry. (**A**) DC maturation phenotype was determined by CD11c and MHC-II staining. Percentage of positive cell for each marker is shown. (**B**) Bi-parametric analysis of CD11c and MHC-II. The gate of cells expressing both CD11c and high levels of MHC-II, is indicated. The expression of MHC-II (**C**) and MHC-I (**D**) is shown. (**E**) Analysis of the maturation marker CD83. (**F**) Percentages of low and high positive cells, and the MFI for the marker. Representative plots (top) and mean ± SEM of percentage of positive cells and GMFI (bottom) are shown. Values of the untreated iDC condition were compared to those of the different treatments using *t* test. *n* = 6 donors for iDC, Spike, and LPS conditions; *n* = 5 donors for RBD. Only significant differences are indicated. * *p* < 0.05, ** *p* < 0.02, *** *p* < 0.01, **** *p* < 0.006.

**Figure 2 cells-10-03279-f002:**
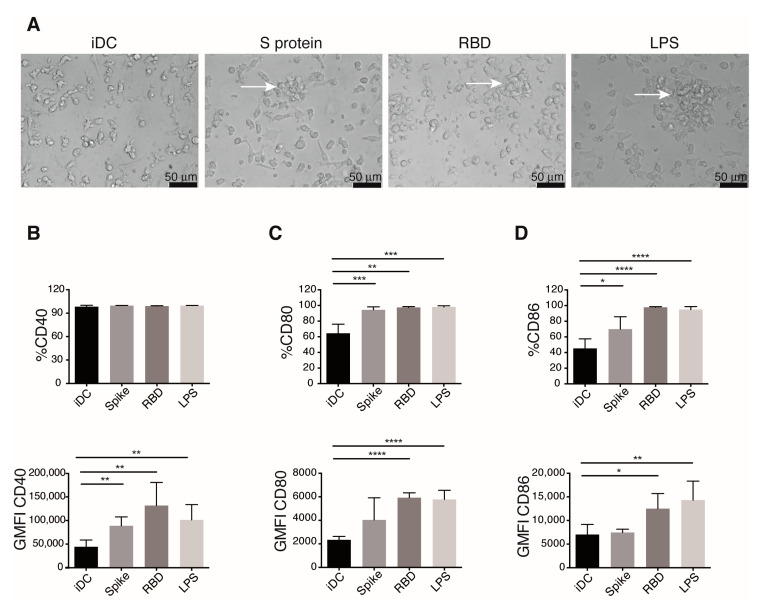
Spike protein and the RBD of SARS-CoV-2 promote iDCs activation. Twenty-four hours after the addition of indicated stimulus DCs, populations were observed by microscope, stained, and analyzed by flow cytometry for the expression of the co-stimulatory molecules. (**A**) Activation in stimulated DCs marked by cells clusters formation (arrows) were documented by bright field microscope photography. Representative images of at least five experiments are shown. Expression of co-stimulatory molecules CD40 (**B**), CD80 (**C**), and CD86 (**D**), was determined. Percentage of positivity (upper) and GMFI (lower) are shown. Mean ± SEM. Values of the untreated iDC condition were compared to those of the different treatments using *t* test. *n*= 6 donors for iDC, Spike, and LPS conditions; *n* = 5 donors for RBD. Only significant differences are indicated. * *p* < 0.05, ** *p* < 0.02, *** *p* < 0.01, **** *p* < 0.006.

**Figure 3 cells-10-03279-f003:**
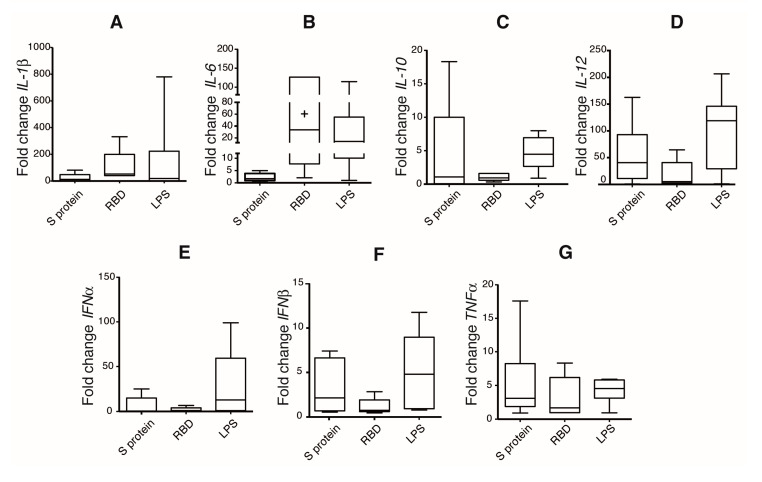
Spike protein and RBD of SARS-CoV-2 induce a proinflammatory activation program in iDCs. Cells were incubated with indicated stimulus for 24 h; relative gene expression of different cytokines was evaluated by real-time PCR and normalized to the basal condition. Relative expression of IL-1β (**A**), IL-6 (**B**), IL-10 (**C**), IL-12 (**D**), IFNα (**E**), IFNβ (**F**), and TNFα (**G**), is shown. *n* = 6 donors for iDC, Spike, and LPS conditions; *n* = 5 donors for RBD.

**Figure 4 cells-10-03279-f004:**
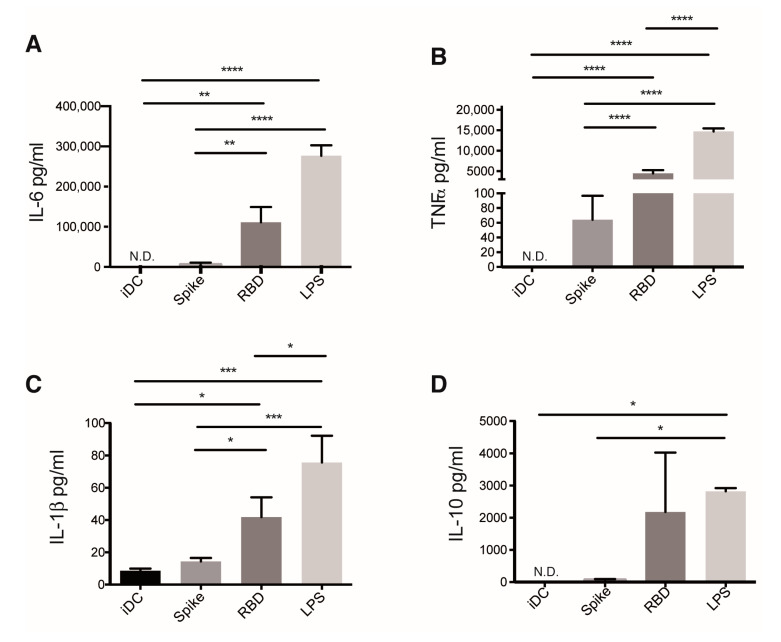
Spike and RBD of SARS-CoV-2 promotes proinflammatory cytokine production by DCs. Cells were incubated with the indicated stimulus, and the concentration of different cytokines secreted after 24 h was measured in supernatants by ELISA. IL-6 (**A**), TNF⍺ (**B**), IL-1β (**C**), and IL-10 (**D**). The graphs show mean ± SEM of two independent experiments; N.D., not detected; multiple comparisons of the values of different treatments by using One-way ANOVA test. *n* = 4 donors in each case. Only significant differences are indicated. * *p* < 0.05, ** *p* < 0.02, *** *p* < 0.01, **** *p* < 0.006.

**Figure 5 cells-10-03279-f005:**
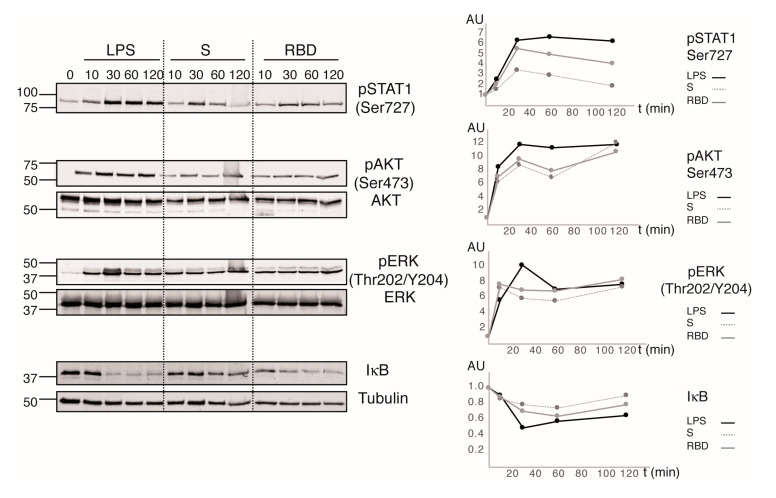
Spike protein and RBD of SARS-CoV-2 triggers AKT, ERK, and STAT1 activation on DCs. Cells were treated with different stimuli for the times indicated. Cells lysates were analyzed by immunoblot using the indicated antibodies. Densitometric analysis was performed using Image J software (v 2.1.0/1.53c). Phosphorylation status of AKT and ERK was normalized to that of the total non-phosphorylated protein, whereas STAT1 phosphorylation and IκB degradation were determined using Tubulin as loading control. Relative values were determined by considering the basal condition = 1 and illustrated in the corresponding graphics in the right. Data shows the results of one donor; additional independent experiments made with the other three donors showed a similar tendency.

**Figure 6 cells-10-03279-f006:**
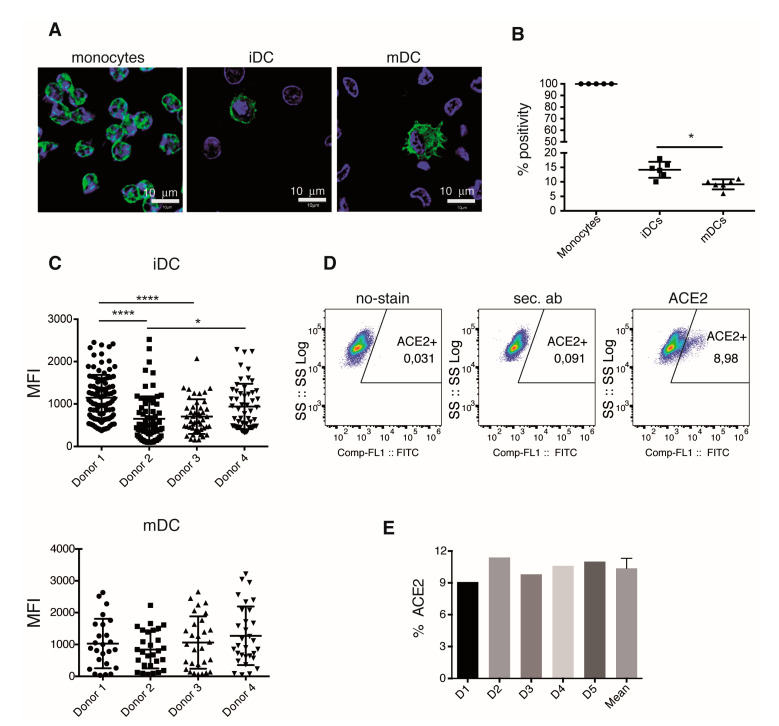
iDCs differentially express ACE2. Human monocytes, iDCs and mDCs, were fixed and analyzed by confocal microscopy to determine ACE2 expression. (**A**) ACE2 (green) and nucleus (blue) in confocal images of one representative experiment. (**B**) Percentage of ACE2 positive cells determined as (**A**) in different donors. (**C**) ACE2 inter-donor expression was evaluated in the ACE2 positive cells from different donors by analyses of MFI of Z stacks images. Comparison of ACE2 expression in iDCs (top) and mDCs (bottom). (**D**) ACE2 expression was determined in iDCs by flow cytometry. Representative plot of one experiment. Profile of unstained control cells (left), secondary ab (middle), and ACE2-stained cells (right). (**E**) Percentage of ACE2 positive iDCs per donor determined by flow cytometry. Statistical comparisons were performed by using One-way ANOVA test. Only significant differences are indicated. * *p* < 0.05, **** *p* < 0.006. In (**B**) and (**C**), *n* = 4, in E, *n* = 5 donors.

**Figure 7 cells-10-03279-f007:**
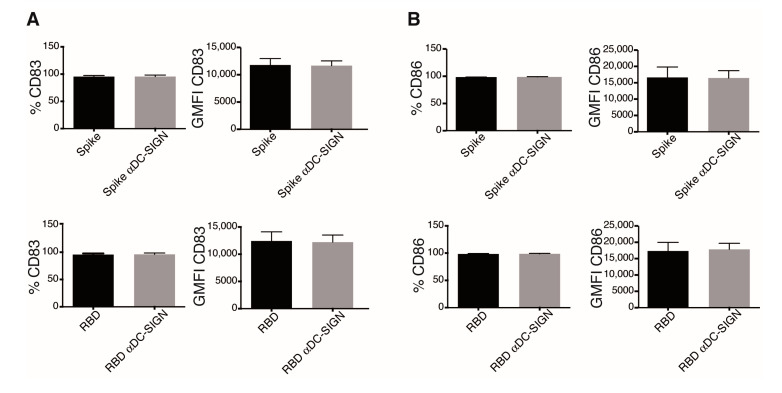
Dendritic cell activation in response to Spike protein and the RBD is DC-SIGN independent. iDCs were preincubated with an anti-DC-SIGN antibody, stimulated as indicated for 24 h, and then stained and analyzed by flow cytometry. (**A**) CD83 and (**B**) CD86 expression in DCs with the indicated treatment are shown. Graphs show percentage of positivity (left) and GMFI (right) of each marker. Graphs show mean ± SEM; *n* = 5 different donors.

## Data Availability

The data presented in this study are available upon request.

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
