# Peer review of "SARS-CoV-2 Spike Protein and Its Receptor Binding Domain Promote a Proinflammatory Activation Profile on Human Dendritic Cells"

_cells, 2021, doi:10.3390/cells10123279_

Round 1

Reviewer 1 Report

The study is interesting. There are some comments and suggestions to improve the manuscript.

  1. In this study, the authors showed the activation of pro-inflammatory pathways after the S protein treatment. In the recently published paper by Colunga etc  "The SARS-CoV-2 spike protein subunit S1 induces COVID-19-like acute lung injury in Κ18-hACE2 transgenic mice and barrier dysfunction in human endothelial cells" the authors demonstrated that the intact S protein does not provoke alveolar inflammation in both in vitro and in vivo, only Subunit 1 of S protein does. The authors should discuss this differences.
  2.  Methods say that author used ANOVA for statistical analysis when comparing more than two groups. However, In figures 1-2, legends say a t-test was used. Please correct the legend or methods. 
  3. Fig. 5: please provide the densitometry and present the results as a graph. 
  4. Please specify the n=? in the methods and figure legends. 

Author Response

Manuscript Cells-1437380

Response to reviewer´s comments

Reviewer #1

We thank the reviewer for analyzing our study and find it interesting. We also appreciate his/her insightful comments. 

In response to the points raised, we address them as follows:

Comment 1. In this study, the authors showed the activation of pro-inflammatory pathways after the S protein treatment. In the recently published paper by Colunga etc. "The SARS-CoV-2 spike protein subunit S1 induces COVID-19-like acute lung injury in Κ18-hACE2 transgenic mice and barrier dysfunction in human endothelial cells" the authors demonstrated that the intact S protein does not provoke alveolar inflammation in both in vitro and in vivo, only Subunit 1 of S protein does. The authors should discuss these differences.

Following the referee’s valuable advice, we have contrasted our findings with those reported at Colunga et al., 2021.

These authors find that the S1 subunit of the Spike protein induce acute lung injury, very likely through triggering massive inflammation. Differently, the Spike protein had practically no effect. As it is described in our work, we found that in general RBD promotes stronger effects than those induced by the Spike protein. Moreover, in some cases RBD effects were similar to those exerted by LPS on dendritic cells. RBD region is contained in the S1 subunit, and therefore our results coincide with that of Colunga and co-authors on the potent inflammatory effect triggered by this region of the Spike protein.

While we observed mild proinflammatory response to S protein challenge, Colunga and co-authors report little or no effect. Such discrepancies could be attributed to the differences in experimental approaches. Our manuscript focused only on human DCs, which express ACE2, but also other molecules that could act as PRRs for PAMPs recognition and might play a role in the effects promoted by the complete Spike protein. The use of the K18-hACE2 mice as a model predicted a main role of the epithelial cells expressing ACE2, which may be more easily targeted by the S1 subunit than by the complete protein. In addition, the use of a complete animal as a model constitutes a more complicated landscape in which many players may control the final inflammatory output. It was nonetheless intriguing that, although at much minor extent, S1 subunit also promoted inflammation in wild type mice. This may be the result of S1 interaction with other cell receptors. Participation of these additional receptors are indeed discussed in our manuscript.

We agree with the referee that discuss the differences, and similitudes, between our findings and those of the paper mentioned is worth, so we now include this comparative in the modified version of our manuscript (Discussion section, paragraph number 4). Since RBD exerted stronger effect than Spike, we also modified the manuscript text to include RBD.

Comment 2. Methods say that author used ANOVA for statistical analysis when comparing more than two groups. However, in figures 1-2, legends say a t-test was used. Please, correct the legend or methods. 

We apologize if our description led to confusion. We have rewritten this part in Methods and indicate the analysis in each figure legend.

 Comment 3. Fig. 5: please provide the densitometry and present the results as a graph. 

The densitometry values were indicated beneath each blot in the first version of our manuscript. For improve clarity we have now present them as a graph to facilitate comparisons.

Comment 4. Please specify the n=? in the methods and figure legends. 

We have rewritten the figure legends to include a more complete description of the number of independent experiments and the number of donors included.

Finally, the manuscript was also checked for correct some fine/minor spell mistakes.  

Reviewer 2 Report

Barrenda et.al. investigate the response of human antigen presenting cells, specifically dendritic cells to SARS-CoV-2 spike protein and showed their response with co-stimulatory molecules expression and pro-inflammatory cytokine profile.

Main questions:

  1. It would be interesting to see respond of T cells to dendritic cells stimulated with SARS-CoV-2 spike protein?
  2. What was the purity of isolated monocytes?
  3. It is not clear how the antibody for flow cytometry were combined since many markers are conjugated with the same color? Please, include gating strategy and how authors determine expression of markers since there was no isotype control included.
  4. Also, there is the question how authors determine live cells? Which marker or method was used to look at that?
  5. Since the is a lot of variability in the cytokine secretion authors should include more donors. It is very difficult to make any conclusion based on three donors, a specially when just one of them respond to treatment e.g. IL-10 cytokine expression. Is it typical response of dendritic cells have a pick of secretion in 12 hours time point? In most of the cytokine expression is it the case. Please comment on that.
  6. It would be very important to know what is the age and sex of PBMCs donors used in the experiments. As mention in the article particularly the eldest people respond with severe symptoms.

Minor comments:

  1. In the main figure e.g. 2 or 7 there is no need to show the same data in two different way. One of them should be transferred to supplementary data.
  2. In Figure 6 C please explain with more details how the MFI Z, stacks were performed from three donors?
  3. In Figure 6D should be used isotype control to determine the expression of ACE2.
  4. Statistical analysis is not consistent. Why in Figure 1 and 2 is used t- test since there is more than one group to compare? Why in Figure 4 are used both T-test and One-way ANOVA?

Author Response

Manuscript Cells-1437380

Response to reviewer comments

Reviewer #2

We thank the reviewer for his/her attention to our manuscript and the questions regarding further experiments and analysis/discussion of our experiments. We also appreciate his/her insightful comments to improve the manuscript.

We have addressed the comments as follows:

Comment 1. It would be interesting to see respond of T cells to dendritic cells stimulated with SARS-CoV-2 spike protein?

     We agree with the referee on the importance of this issue.  Determine the ability of hyperinflammatory DCs, derived from RBD challenge, to properly activate T cells would be worth to get a glimpse of the mechanisms developed by the adaptive immune response when the innate immune response is dysregulated in the onset of SARS-CoV-2 infection. We are now actively designing T cell-based experiments, but we believe that these studies merit a detailed analysis that is beyond the scope of the present study.

Comment 2. What was the purity of isolated monocytes?

After purification we labelled the cells with a CD14+ antibody followed by flow cytometry analysis. The purity was at least 96% for every experiment. We modified the Methods section to introduce this information and now we include a representative histogram the Supplemental Figure 1A.  

Comment 3. It is not clear how the antibody for flow cytometry were combined since many markers are conjugated with the same color? Please, include gating strategy and how authors determine expression of markers since there was no isotype control included.

We apologize if our description was incomplete and lead to confusion. We stained the analyzed molecules in three different batches combining different fluorophore-conjugated antibodies: batch 1, included CD11c-PE and HLA-DR-FITC; batch 2, included CD80-APC, CD86-PE and CD83-FITC, and batch 3, included CD40-PE and HLA-ABC-FITC. We have rewritten this part in the Methods section and included a diagram of the staining flow strategy in the new supplementary figure 1B.

Regarding controls, we determine the expression of every molecule by using either unstained cells, or fluorescence minus one (FMO) control. This section was expanded in the Methods description and representative histograms of single stains are now included in the Supplemental Figure 2. 

Comment 4. Also, there is the question how authors determine live cells? Which marker or method was used to look at that?

Our cultures usually have no a big number of dead cells, we routinely determined cell dead index by using Trypan Blue Exclusion and a Life Countess ii (ThermoFisher Scientific).

Prior to analysis, the harvested cultures were centrifugated at low speed (800 rpm 10 min) to discard debris, then the pellet was suspended in a determined volume and the percentage of live cells determined by Trypan Blue Exclusion. Then in flow cytometer experiments, DCs population was identified according to its FSC and SSC parameter from single events, which further discard a large proportion of dead cells. We modified the Methods section to introduce this information and representative histograms of the gating are included in the Figure S1B, together with the staining ab combination.

Comment 5. Since the is a lot of variability in the cytokine secretion authors should include more donors. It is very difficult to make any conclusion based on three donors, a specially when just one of them respond to treatment e.g. IL-10 cytokine expression.

We apologize if our description of the procedures was no accurate and lead to misinterpretation. Cytokine-secretion data shown are indeed from 3 donors, and here neither Spike nor RBD proteins had a significative effect on IL-10 secretion. We obtained similar results when performed another independent experiment that included a different set of donors. In both experiments LPS treatment indicated that cells were capable of secreting this cytokine, but responses to the viral proteins exhibited the same variation.

A different set of donors were also used in the experiments of mRNA quantification. Here IL-10 expression analysis showed high variability, but in general it coincides with that observed in IL-10 secretion since in RBD and Spike the expression range was much minor than that of LPS.

Variability is inherent to our experimental approach, since highly variable responses among individuals were expected and are well documented (e.g. Wurfel, MM et al., 2005, J. Immunol. 175 (4): 2570; Giamarellos-Bourboulis EJ et al., 2020, Cell Host Microbe, 27(6): 992).  Nonetheless, data from at least eight donors show that the Spike and RBD promote strong proinflammatory response by DCs if we consider the rest of the cytokines analyzed.  

Is it typical response of dendritic cells have a pick of secretion in 12 hours time point? In most of the cytokine expression is it the case. Please comment on that.

We decided to analyze cytokine secretion gene expression at 12 and 24 h since, in our laboratory, we have analyzed gene expression of different cytokines in distinct cell lines and identified peaks of expression ranging from 10 h (for monocyte-like cells) to 24 h (for lymphocytes). Our data suggest that in most of the cytokines analyzed the expression decreases at 24 h; in agreement Ebner et al., (J Immunol 2001, 166: 633-641) reported that once DC reached a maturation state, the production of IL-12 is reduced.

Once the cytokine gene is transcribed, it is expected that translation and secretion lead to cytokine accumulation in the media. Analysis of the supernatants indeed showed that IL-6 increased over time (12 h, in Supplemental Figure 3, vs 24 h, in Figure 4). Differently we noticed a slight diminution in the amount of TNFalpha at 24h in the cells treated with either Spike or RBD proteins, whereas this cytokine accumulates after LPS stimulation. This suggests self-consumption of the cytokine.

Comment 6. It would be very important to know what is the age and sex of PBMCs donors used in the experiments. As mention in the article particularly the eldest people respond with severe symptoms.

     We agree with the referee in this observation. As indicated in the Methods section we obtain the biological material (buffy coats) from a blood bank in which no personal data were register to assure compliance with ethical standards.

We are now actively working in planning a system with cohorts for age and sex to assess the possibles link between the magnitude of the inflammatory response elicited by the viral proteins used here and individual´s features like age, sex and other factors, like mild pathologies. We think that all these studies merit a detailed analysis that is beyond the scope of the present study.

Minor comments:

In the main figure e.g. 2 or 7 there is no need to show the same data in two different way. One of them should be transferred to supplementary data.

We partly agree with the reviewer about this comment. In Figure 2 the graphs are intended to give different information. We are depicting the number of cells expressing the marker (%) and the signal intensity on a per cell basis (GMFI). Comparison of the graphics indicates different tendencies.

Nonetheless we agree with the referee in the case of Figure 7, and followed his/her advice. Data were simplified and part was moved to the supplemental section, in the new Supplemental figure 5.

In Figure 6 C please explain with more details how the MFI Z, stacks were performed from three donors?

We have modified the description in Methods section for improve clarity. Cells from at least three donors were stained and at minimum of thirty cells for donor and condition were captured.

For ACE2 quantification the Z stacks images were acquired as follows. Ten sequential planes were acquired at axial (z) spacing of 0.5 μm to form a z-stack. The using ImageJ software channel corresponding to ACE2 signal were flattened along the z axis as maximum intensity projections representing a “top” (x-y) view of the volume. Region of interest (ROI) was set for every individual cell and mean fluorescence was measured using the ROI Manager tool.

In Figure 6D should be used isotype control to determine the expression of ACE2.

Since no ACE2 antibody for flow cytometry were available that matched the Ig subtype, we used unstained cells and the secondary ab to identify ACE2 ab specific staining range. We modify the figure 6D to include the secondary ab control.

Statistical analysis is not consistent. Why in Figure 1 and 2 is used t- test since there is more than one group to compare? Why in Figure 4 are used both T-test and One-way ANOVA?

We apologize if our description led to confusion. We have rewritten this part in Methods and indicated the method used in each case in the corresponding figure legend. 

English language and style are fine/minor spell check required.

The manuscript was checked for correct some fine/minor spell mistakes.  

Round 2

Reviewer 2 Report

Authors replay to all comments and I have no more comments

Author Response

We are glad to learn that the referees found our responses satisfactory and appreciate your careful revision of the manuscript.

We have modified the manuscript to fulfill in the most satisfactory way the raised points considered by the editor.

If required, please see the "Point by point response” to the editor.